# Acupuncture for Post-Operative Pain Relief and Functional Improvement in Tibial Fracture: A Systematic Review and Meta-Analysis

**DOI:** 10.3390/healthcare13222883

**Published:** 2025-11-12

**Authors:** Whisung Cho, Woo-Chul Shin, Hyungsuk Kim, Won-Seok Chung, Mi-Yeon Song, Yousuk Youn, Jae-Heung Cho

**Affiliations:** 1Department of Korean Rehabilitation Medicine, College of Korean Medicine, Kyung Hee University Korean Medicine Hospital, Kyung Hee University, Seoul 02447, Republic of Korea; whisung78@naver.com (W.C.); eddyshin42@naver.com (W.-C.S.); kim0874@hanmail.net (H.K.); omdluke@naver.com (W.-S.C.); mysong@khu.ac.kr (M.-Y.S.); 2Chamjalham Hospital of Korean Medicine, Suwon 16263, Republic of Korea; spineyoun@gmail.com

**Keywords:** acupuncture, tibial fracture, systematic review, meta-analysis

## Abstract

**Background/Objectives**: Acupuncture has been used to manage various fractures. The aim of this study is to evaluate the effectiveness of acupuncture in reducing post-operative pain and improving function after tibial fracture surgery. **Methods**: A systematic review and meta-analysis of randomized controlled trials (RCTs) was conducted to evaluate acupuncture as an intervention for tibial fractures. Eight databases were searched, covering studies from inception to August 2024. **Results**: Sixteen RCTs with 1315 patients were analyzed. Acupuncture, when combined with conventional rehabilitation (CR) or medication, significantly improved pain (MD −1.03, 95% CI [−1.44, −0.62]), Hospital for Special Surgery (HSS) score (MD 13.21, 95% CI [9.16, 17.26]), and range of motion (ROM) of knee joint (SMD 1.80, 95% CI [0.32, 3.28]) compared to CR or medication alone. It showed significant effects on effective rate (OR 4.92, 95% CI [2.79, 8.68]; I^2^ = 0%) and showed a lower incidence of complications (OR 0.13, 95% CI [0.06, 0.26]). **Conclusions**: Acupuncture combined with CR or medication during the rehabilitation period after tibial fracture surgery can reduce pain, improve knee joint function, increase knee ROM, and decrease post-operative complications, compared to CR or medication alone.

## 1. Introduction

Tibial fractures are broadly divided into shaft, distal and proximal fractures [1]. Tibial shaft fractures account for approximately 37% of all long bone fractures in adults, with an overall incidence of 17–21 per 100,000 people [2]. They have the highest incidence in active males aged 10–20 years and are often caused by high-energy trauma such as car accidents, sports, and falls from heights [3]. Distal tibial fractures are also often caused by high-energy injury, with a reported annual incidence of 9.1 per 100,000 people [4]. Proximal tibial fractures occur between the ages of 40 and 60 by high-energy trauma such as car accidents, with a reported annual incidence of 10 per 100,000 people [5]. Intra-articular fractures of the proximal tibia are commonly referred to as tibial plateau fractures, which account for approximately 1% of all fractures [6,7].

The treatment of tibial fractures is divided into conservative treatment, such as cast or surgical treatment, depending on the location and type of the fracture. Due to the prolonged immobilization and high risk of malunion associated with conservative treatment, surgical treatments like intramedullary nailing are commonly performed [1,4,8,9].

After achieving fracture reduction and stabilization, medication and rehabilitation therapy are administered for post-operative care, but reducing pain, improving lower limb function, and shortening the fracture healing time after surgery are challenging [10]. The analgesics, anti-inflammatory, and anti-edematous drugs used in post-surgical care have the advantage of fast-acting and long-lasting effects, but they come with many side effects and require close monitoring [9].

Acupuncture, a traditional therapeutic modality, has been widely applied for the management of musculoskeletal pain and functional impairment [11,12,13]. Its mechanisms are thought to involve modulation of endogenous opioid peptides, serotonin and norepinephrine in the central nervous system, along with improved local blood circulation and anti-inflammatory effects at the site of injury [14,15]. Experimental studies further suggest that acupuncture may promote tissue repair and functional recovery by regulating growth factor expression [16]. Clinically, a previous systematic review has reported that acupuncture reduces post-operative pain and opioid consumption after total knee arthroplasty (TKA) [17].

Building on the evidence, acupuncture has shown analgesic effects and functional improvement in patients with tibial fractures [9,18,19]. As it can serve as an alternative treatment to avoid the adverse effects of typical medication and enhance rehabilitation outcomes by improving function, acupuncture may represent an important therapeutic option for tibial fractures. In addition, a recent meta-analysis suggested that acupuncture can reduce pain after humeral fractures [20]. However, most studies on the effects of acupuncture on fractures have been composed of case studies and randomized controlled trials (RCTs), with few meta-analyses, and none specifically focused on tibial fractures. Accordingly, this study was undertaken to fill this gap by providing the first systematic review and meta-analysis on acupuncture after tibial fracture surgery, with the aim of evaluating its clinical effectiveness and safety, thereby offering evidence that complements and extends previous reviews in related contexts.

## 2. Materials and Methods

The protocol of this study has been registered in the PROSPERO International prospective register of systematic reviews (CRD42025578048; registered on 31 January 2025). This systematic review follows the Preferred Reporting Items for Systematic Reviews and Meta-Analysis (PRISMA) guidelines (Appendix A) [21,22].

### 2.1. Criteria for Inclusion and Exclusion

#### 2.1.1. Study Types

This study included only randomized controlled trials (RCTs). Studies were included regardless of their publication language.

#### 2.1.2. Participants

Patients who were treated with acupuncture after tibial fracture surgery were included. There were no restrictions on sex, age, and the type of procedure used during surgery. We excluded patients with knee fractures of uncertain location.

#### 2.1.3. Interventions

Pharmacological treatments, physiotherapy, fumigation, washing and patient education are defined as “conventional rehabilitation (CR)”. Physiotherapy included rehabilitation programmes such as PNF training, mobilization of knee joint using continuous passive motion (CPM), partial weight bearing exercise, and physical exercise. Pharmacological treatments included various analgesics administered orally or via intravenous injection. Fumigation and washing are performed by soaking the surgical site in a decoction of medicinal herbs and then washing it.

For the experimental group, the intervention was acupuncture in addition to the CR provided to the control group. All types of acupuncture interventions were included in this review, encompassing both invasive techniques such as manual acupuncture, electroacupuncture, auricular acupuncture, and non-invasive methods such as transcutaneous electrical acupoint stimulation (TEAS) and laser acupuncture. Warm needling was excluded from this review as it is an intervention performed in combination with moxibustion, in order to isolate the effects of acupuncture alone.

For the control group, the intervention was CR. The control group differed from the experimental group only in that it did not receive acupuncture treatment, and cases where the control group included acupuncture as intervention were excluded.

#### 2.1.4. Outcomes

For evaluating pain, we used the visual analogue scale (VAS). For evaluating knee joint functions, Hospital for Special Surgery (HSS) score and the range of motion (ROM) of knee joint were used. HSS score consists of pain, function, range of motion, muscle strength, flexion deformity, and instability, and the total score is 100 points. We used the effective rate, a frequently selected estimating measure. The effective rate was calculated as the ratio of patients who showed complete or partial improvement and patients who showed no improvement. Incidence of complications after surgery and adverse events were also used for our study. Other outcomes were excluded due to their low frequency of use.

### 2.2. Search Strategy

Cochrane library, PubMed, EMBASE, China National Knowledge Infrastructure (CNKI), Japan Science and Technology Information Aggregator Electronic (J-stage), Research Information Service System (RISS), Korean Studies Information Service System (KISS), and Oriental Medicine Advanced Searching Integrated System (OASIS) were searched for articles. The search was performed by two reviewers (W.C. and H.K.) from database inception to 31 August 2024. There were no restrictions on language or the year of publication. The search process was based on three keywords: (1) “tibial fracture”, (2) “acupuncture”, (3) “randomized controlled trial”. The complete search strategy is presented in Appendix A.

### 2.3. Article Screening

With predetermined search strategies, two reviewers (W.C. and H.K.) independently searched the databases. Duplicate records were removed using EndNote (version 21, Clarivate, Philadelphia, PA, USA). The reviewers independently selected studies by predetermined inclusion and exclusion criteria based on a review of the title, abstract, and main text. JH Cho made the final decision when consensus on the selection process could not be reached.

### 2.4. Data Extraction

Data were extracted from each article through a full-text review by two reviewers (W.-C.S. and M.-Y.S.). Details of intervention in experimental groups and control groups, types of outcome measurements, duration of intervention, adverse events, and acupuncture points were obtained and organized using Microsoft Excel for Microsoft 365 (Microsoft Corp., Redmond, WA, USA). To identify frequently used acupoints, the frequency of acupoint usage was analyzed. JH Cho made the final decision when consensus could not be reached.

### 2.5. Data Collection and Analysis

Two reviewers (W.-C.S. and M.-Y.S.) reviewed the main text of the selected studies to confirm studies to include in the analysis. They summarized in a table for information such as authors, year of publication, sample size, details of intervention in experimental groups and control groups, outcomes.

We conducted meta-analysis using Cochrane Collaboration software [Review Manager Version 5.4 for Windows. Copenhagen: The Nordic Cochrane Centre].

Continuous data such as visual analogue scale (VAS), Hospital for Special Surgery (HSS) score, range of motion (ROM) was expressed in terms of the mean difference (MD) or standardized mean difference (SMD) with 95% confidence interval (CI) using inverse variance estimation. Dichotomous data such as effective rate and incidence of complications after surgery was expressed in terms of the odds ratio (OR) and 95% CI using the Mantel–Haenszel estimation method. A chi-square test (test with *p*-value of *p* < 0.10) and Higgins I^2^ statistic were used to analyze the heterogeneity. If I^2^ is below 50%, fixed effect model was used in analysis. If I^2^ is over 50%, random-effects model was used in analyses. Subgroup analysis was performed to analyze the cause of heterogeneity when significant heterogeneity was observed. We did not conduct a meta-analysis if meaningful heterogeneity could not be explained by subgroup analysis. Finally, a leave-one-out sensitivity analysis was undertaken to evaluate robustness, sequentially omitting each study and re-running the meta-analysis at each iteration.

### 2.6. Methodological Quality Assessment for the Included Studies

#### 2.6.1. Risk of Bias

Two independent researchers (W.C. and W.-S.C.) evaluated the risk of bias of the selected studies using the Cochrane Collaboration tool (risk of bias, ROB). The risk of bias was assessed in seven domains (random sequence generation, allocation concealment, blinding of participants and personnel, blinding of outcome assessment, incomplete outcome data, selective outcome reporting, and other sources of bias) with rated each area as high, low or unclear. When consensus on the assessment could not be reached, JH Cho made the final decision.

#### 2.6.2. The Level of Evidence

The level of evidence was analyzed with reference to The Grading of Recommendations, Assessment, Development and Evaluation (GRADE) [23]. Risk of bias, inconsistency, indirectness, imprecision, publication bias, large effect, plausible confounding, and dose response gradient were assessed. Assessments were determined as one of four grades: high, moderate, low, or very low.

## 3. Results

### 3.1. Description of Studies

A total of 111 studies were retrieved from the online search. We first excluded 10 duplicate studies. The title and abstracts of the remaining studies were further examined, and 63 studies were eliminated for several reasons: patients who did not have a tibial fracture or who did not undergo surgery after a tibial fracture, interventions that did not meet the inclusion criteria, animal studies, not in the form of academic papers. After a detailed review of full text, 22 studies were eliminated for several reasons: not RCTs, acupuncture treatment in the control group, mixed treatments. Finally, 16 studies [9,18,19,24,25,26,27,28,29,30,31,32,33,34,35,36] were included for our analysis (Figure 1).

Quantitative and qualitative analyses were conducted on 16 studies. All patients in the studies underwent surgery after tibial fracture, and 12 studies included only patients with tibial plateau fracture. Only four studies [9,29,33,36] presented the type of surgery, the remainder did not present the type of surgery. The details of fracture type of participants are summarized in Appendix A.

All selected studies used manual acupuncture or TEAS or electroacupuncture combined with the control group treatment, while the control group treated with basic rehabilitation or medication or TCM fumigation. A total 12 studies used manual acupuncture; one study used TEAS and three studies used electroacupuncture as the intervention. The most commonly used acupoint was GB34, followed by EX-LE5, SP9, ST34, ST37, BL40, EX-LE2, and SP10. These frequently used acupuncture points are located around the knee joint. These frequently used acupuncture points are organized in Table 1. Most acupoints were on the fracture side, but there were also cases where acupoints on the opposite side were used [19,27,28,33,35].

Among selected studies, five studies used VAS, 7 studies used HSS score, four studies used ROM of knee joint, seven studies used effective rate as an end point. A total of eight studies reported incidence of complications after surgery. The details are summarized in Table 2.

### 3.2. Risk of Bias Assessment

The assessment of risk of bias is summarized in Figure 2.

#### 3.2.1. Random Sequence Generation

There were eight studies [11,12,18,19,20,22,23,25] that showed a low risk of bias because they used a random number table or computer-generated random sequence. Three studies [21,26,28] used an inappropriate randomization method, while five studies [9,24,27,29,30] did not present specific descriptions about their randomization sequence.

#### 3.2.2. Allocation Concealment

Only one study [11] adequately carried out allocation using sealed envelope. The other studies [9,12,18,19,20,21,22,23,24,25,26,27,28,29,30] were judged to show unclear risk because they did not present the method of allocation concealment.

#### 3.2.3. Blinding of Participants and Personnel

Due to the nature of RCTs on acupuncture, all 16 studies were judged to have a high risk of bias.

#### 3.2.4. Blinding of Outcome Assessment

Only one study [11] reported that evaluators were independent of the intervention and blinded to the assigned groups. The other studies [9,12,18,19,20,21,22,23,24,25,26,27,28,29,30] were judged to show unclear risk because they did not clarify the specific methods used to prevent detection bias.

#### 3.2.5. Incomplete Outcome Data

One study [11] reported dropouts, but this was judged to be insignificant because the dropout rates of the groups were very low. The other studies [9,12,18,19,20,21,22,23,24,25,26,27,28,29,30] were judged to show a low risk of bias because there was no missing data.

#### 3.2.6. Selective Reporting

Two studies [26,28] were judged to show a high risk of bias because specific HSS scores, one of the primary outcomes, were not reported as mean or standard deviation. Also, one study [25] was judged to show a high risk of bias because the total HSS score and muscle strength score of HSS were missing. Another study [23] was judged to show a high risk of bias because the total HSS scores were missing and there were no reports of adverse events. One study [21] was judged to show unclear risk because it was only VAS-checked after 7 days of treatment, while other outcomes were checked after 8 weeks of treatment and there were no reports of adverse events. A total of five studies [11,12,22,27,29] were judged to show unclear risk because there were no reports of adverse events. The other studies [9,18,19,20,24,30] were judged to show a low risk of bias.

#### 3.2.7. Other Bias

One study [23] was judged to show unclear risk because the method of measuring joint range of motion was not clearly presented. As the total period of acupuncture treatment was not presented, two studies [26,28] were judged to show unclear risk. Another study [9] was judged to show unclear risk because needling duration for each treatment was not presented. Six studies [21,25,27,28,29,30] were judged to show unclear risk because treatment frequency was not presented. The other studies [11,12,18,19,20,22,24] were judged to show a low risk of bias.

### 3.3. Effects of Interventions

The meta-analysis was performed on 16 studies [9,11,12,18,19,20,21,22,23,24,25,26,27,28,29,30]. Five outcomes were analyzed: VAS, HSS score, ROM of knee joint, effective rate, incidence of complications after surgery. SMD was adopted for the meta-analysis of ROM of knee joint, because there were differences between studies in the methods used to measure ROM of knee joint. Because of considerable heterogeneities, a subgroup analysis was conducted for VAS, HSS score, and ROM of knee joint. No meaningful subgroup analyses that could explain the considerable heterogeneity in ROM of knee joint were found.

#### 3.3.1. VAS

In the meta-analysis of five studies [9,11,12,19,21] involving 341 patients, compared to control group, the acupuncture group showed significant improvement in the VAS (MD −1.03, 95% CI [−1.44, −0.62]; I^2^ = 84%; Figure 3), with substantial heterogeneity. The subgroup analysis was based on duration of treatment prior to evaluation. All duration of treatment prior to evaluation showed more significant improvement: 0–1 week (MD −1.21, 95% CI [−2.05, −0.36]; I^2^ = 86%), 6–12 weeks (MD −0.88, 95% CI [−1.33, −0.43]). The sensitivity analysis (leave-one-out method) revealed MD ranging from 0.85 to 1.19 with no loss of significance (Appendix A).

#### 3.3.2. HSS Score

In the meta-analysis of seven studies [19,24,25,30,33,35,36] involving 564 patients, compared to control group, the acupuncture group showed more significant improvement in the total HSS score (MD 13.21, 95% CI [9.16, 17.26]; I^2^ = 94%; Figure 4), with substantial heterogeneity. The subgroup analysis was based on the types of acupuncture treatments. All types of acupuncture treatments showed significantly better improvement: manual acupuncture (MD 10.76, 95% CI [6.97, 14.55]; I^2^ = 92%), electroacupuncture (MD 19.46, 95% CI [17.36, 21.55]; I^2^ = 0%). The sensitivity analysis (leave-one-out method) revealed MD ranging from 12.17 to 14.45 with no loss of significance (Appendix A).

#### 3.3.3. ROM of Knee Joint

In the meta-analysis of four studies [26,28,29,30] involving 368 patients, acupuncture group showed significantly better improvement in the ROM of knee joint compared to control group (SMD 1.80, 95% CI [0.32, 3.28]; I^2^ = 97%; Figure 5), with substantial heterogeneity. The sensitivity analysis (leave-one-out method) revealed SMD ranging from 1.08 to 2.26 (*p* = 0.08 to *p* = 0.008) (Appendix A).

#### 3.3.4. Effective Rate

We used seven studies [19,25,26,28,30,33,35] involving 636 patients in a meta-analysis of the effective rate. The acupuncture group was more effective than the control group (OR 4.92, 95% CI [2.79, 8.68]; I^2^ = 0%; Figure 6). The sensitivity analysis (leave-one-out method) revealed OR ranging from 4.58 to 5.58 with no loss of significance (Appendix A).

#### 3.3.5. Incidence of Complications After Surgery

We used eight studies [24,25,26,30,31,32,34,36] involving 644 patients in a meta-analysis of incidence of complications after surgery. The acupuncture group showed a significantly lower risk of complications compared with the control group (OR 0.13, 95% CI [0.06, 0.26]; I^2^ = 0%; Figure 7). A total of four studies reported specific complications, including arthritis, venous thrombosis, joint stiffness, malunion and nonunion (Appendix A). The sensitivity analysis (leave-one-out method) revealed OR ranging from 0.11 to 0.14 with no loss of significance (Appendix A).

### 3.4. Adverse Events After Acupuncture

One study [9] reported mild adverse events, including one case of dizziness, one case of bleeding at acupuncture site and one case of anorexia after acupuncture.

### 3.5. The Level of Evidence

The assessment of the quality of evidence is summarized in Table 3. The quality of evidence for all outcomes was downgraded by one level because all 16 RCTs showed a high risk of performance bias. VAS and HSS scores, which showed significant improvement, showed considerable heterogeneity, but subgroup analysis showed improvement in heterogeneity, and the direction of effect was consistent, so the quality of evidence for inconsistency was downgraded by one level. Although there was improvement in ROM of knee joint, the quality of evidence for inconsistency was downgraded by two levels due to considerable heterogeneity not explained by subgroup analysis. There was no evidence of indirectness, as all studies directly compared interventions. There was no evidence of imprecision in HSS score, incidence of complications after surgery and effective rate, as the population size was adequate. In the case of VAS and ROM of the knee joint, the sample size significantly exceeded the Optimal Information Size (OIS), approaching 400, and the confidence intervals did not include the null effect; therefore, the grade for imprecision was not downgraded.

## 4. Discussion

Our meta-analysis found that, compared with CR or medication alone, acupuncture combined with CR or medication reduced pain (VAS), improved function, increased ROM and reduced post-operative complications. For pain, a meta-analysis of five RCTs reporting VAS results showed greater pain reduction with acupuncture, and this effect was observed regardless of treatment duration in subgroup analyses. Similarly, a meta-analysis of seven RCTs reporting HSS scores demonstrated that acupuncture combined with CR was more effective than CR alone, with subgroup analyses indicating consistent benefits regardless of the type of acupuncture. In terms of ROM, four RCTs indicated that acupuncture improved knee joint mobility, although the high heterogeneity could not be explained by subgroup analyses. Regarding the effective rate, sevenf RCTs consistently showed that acupuncture combined with CR was more effective than CR alone, with relatively low heterogeneity. However, the use of “effective rate” as an outcome is common in Chinese studies but is not widely recognized internationally, which may affect the validity and comparability of the results.

After fracture, reduction and stabilization are important, but subsequent management and rehabilitation are essential. Early mobilization can help reduce complications such as knee joint stiffness, muscle and bone atrophy, synovial adhesions, and capsular contractions, and also improves range of motion (ROM) and muscle strength [10,37,38]. Although several rehabilitation protocols after tibial fracture have been proposed, controversy remains regarding the most appropriate methods, and there is a growing demand for programmes that alleviate symptoms and improve joint function [39,40].

Acupuncture has been reported to provide analgesic effects and functional improvement by directly stimulating muscles around the joint, making it a potential adjunctive rehabilitation option [9,18,19]. For example, one previous systematic review demonstrated that acupuncture in addition to conventional rehabilitation reduced pain after proximal humerus fracture [20].

The most commonly used acupuncture point was GB34, followed by EX-LE5, SP9, ST34, ST37, BL40, EX-LE2, and SP10. These points are mainly located around the knee joint and are thought to enhance bone healing, reduce pain, and improve function. Evidence from animal models has shown that electroacupuncture near the surgical site of tibial fracture and acupuncture near the surgical site of femur fracture, as well as at distal acupoints, promoted callus formation and bone mineralization during the bone healing process [41,42]. In addition, Li H et al. reported that acupuncture at SP10 alleviated post-operative pain in elderly patients with intertrochanteric fracture [43], and Lian X et al. demonstrated in an animal model that electroacupuncture at SP9 restored injured gastrocnemius muscle and increased growth factor expression, potentially contributing to improved knee and ankle joint function after tibial fracture [44]. However, because only a limited number of studies reported side effects, further high-quality trials are needed to confirm the safety profile.

Complications such as compartment syndrome, knee pain, knee stiffness, ankylosis, deep infection, post-traumatic arthritis, malunion, and nonunion are common concerns after tibial fracture surgery [45,46]. A meta-analysis of eight RCTs that reported complication outcomes demonstrated that acupuncture combined with conventional rehabilitation (CR) was associated with a lower incidence of post-operative complications compared with CR alone. The spectrum of complications in the acupuncture groups mirrored that in the control groups, suggesting that these events were not directly caused by acupuncture treatment. Across trials, acupuncture was generally initiated without a post-operative rest period, and no severe complications, such as deep infection, were reported. One RCT [9] reported only mild, self-limiting adverse events (e.g., dizziness, minor bleeding at the needling site, anorexia), with no serious or life-threatening events. Collectively, these findings suggest that acupuncture may have a favourable safety profile as an adjunct after tibial fracture surgery. Nevertheless, adverse-event reporting was limited and heterogeneous across studies; well-designed trials with standardized safety monitoring are needed to confirm these safety findings.

This study has several limitations. First, among the 16 included trials, 15 were conducted in China and one in Vietnam, which may limit generalizability. Second, allocation concealment and blinding were often poorly reported, which reduces confidence in the findings. In addition, important details on treatment regimen and outcome assessment were frequently missing, which further limits the reliability of the evidence. Third, blinding was not feasible due to the specific nature of acupuncture, introducing a risk of performance bias that may have affected the level of the evidence (Table 3). Fourth, only two trials [33,35] reported bone healing time, and none evaluated long-term functional recovery, which limits the comprehensiveness of the findings. Given the very limited data, a meta-analysis was not feasible. Finally, between-study heterogeneity was substantial for several outcomes—particularly ROM (I^2^ = 97%) and HSS (I^2^ = 94%), and high for VAS (I^2^ = 84%)—which tempers confidence in the precision and generalizability of these estimates. Potential sources include clinical variability (acupoint selection and dosing, timing relative to surgery, type of conventional rehabilitation, follow-up time points) and methodological differences (outcome operationalization/measurement and risk of bias profiles). Subgroup analysis only partially explained the observed heterogeneity, which weakens confidence. Leave-one-out sensitivity analyses supported robustness to single-study influence across outcomes (i.e., pooled effects and significance were largely preserved when omitting individual trials), but this did not resolve the underlying inconsistency. These issues should be considered when interpreting the certainty of the evidence.

## 5. Conclusions

Acupuncture combined with CR or acupuncture combined with medication during the rehabilitation period after tibial fracture surgery may offer potential benefits in reducing pain, enhancing knee joint function, increasing ROM, and decreasing post-operative complications. However, these findings should be interpreted cautiously due to the relatively low methodological quality of the included RCTs, substantial heterogeneity in the meta-analysis, and the overall low certainty of evidence. This study is the first systematic review evaluating the effects of acupuncture after tibial fracture surgery. We expect that our data may serve as a valuable basis for acupuncture research and clinical practice for various fractures, and more high-quality RCT studies in various countries are needed.

## Figures and Tables

**Figure 1 healthcare-13-02883-f001:**
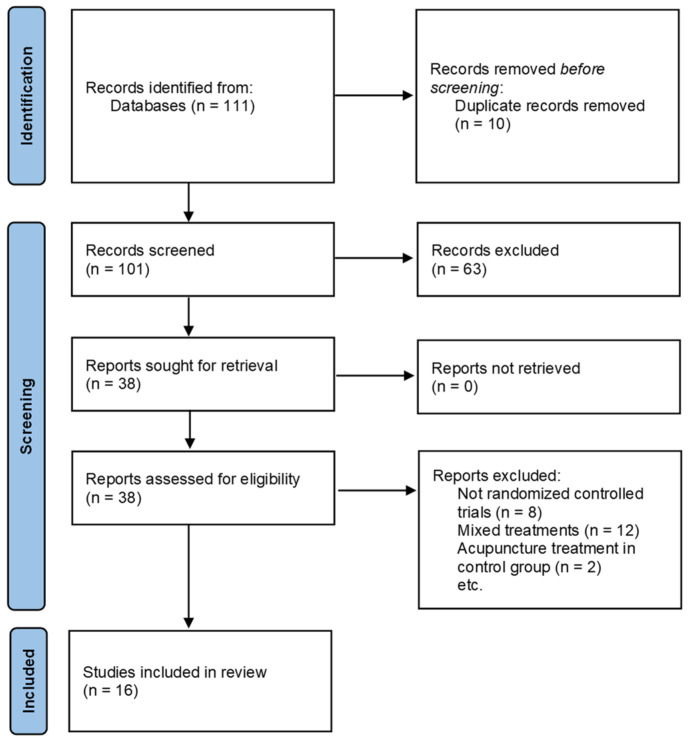
Flow diagram of the study selection process.

**Figure 2 healthcare-13-02883-f002:**
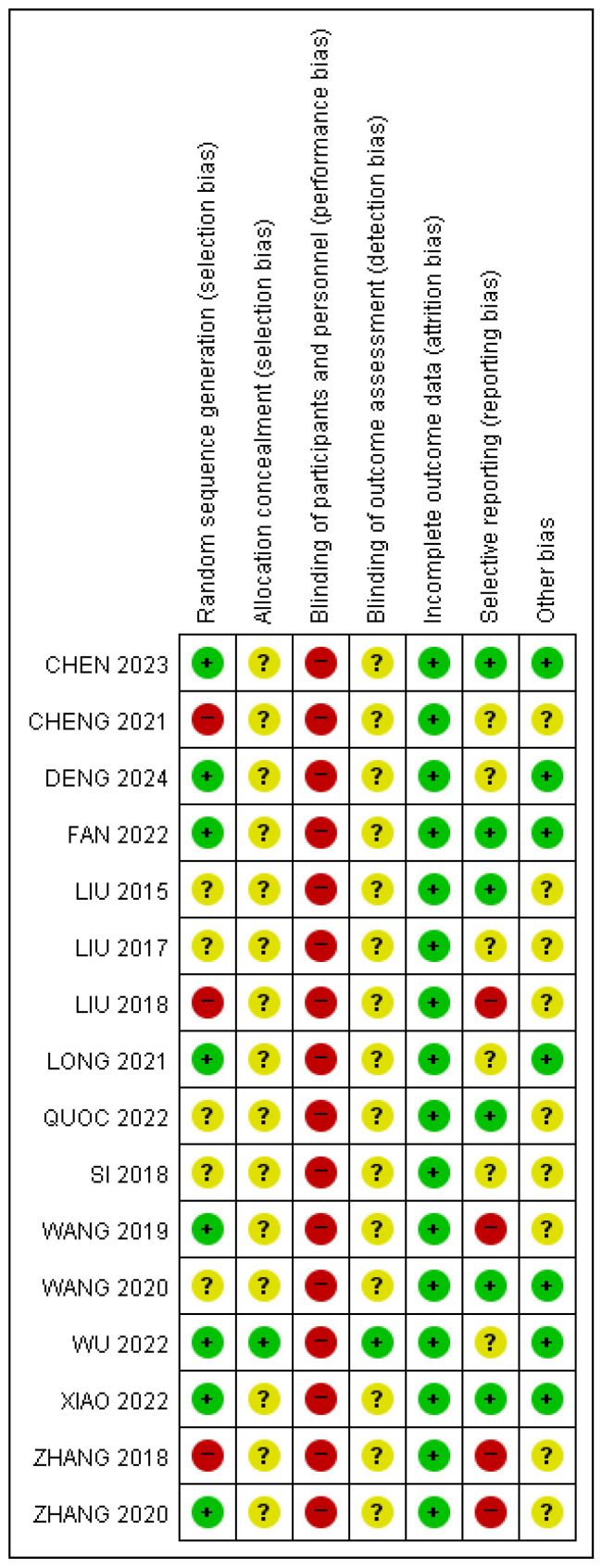
Assessment of the risk of bias of the included randomized controlled trials [9,18,19,24,25,26,27,28,29,30,31,32,33,34,35,36]. Red, yellow, and green circles indicate high, unclear, and low risks of bias, respectively.

**Figure 3 healthcare-13-02883-f003:**
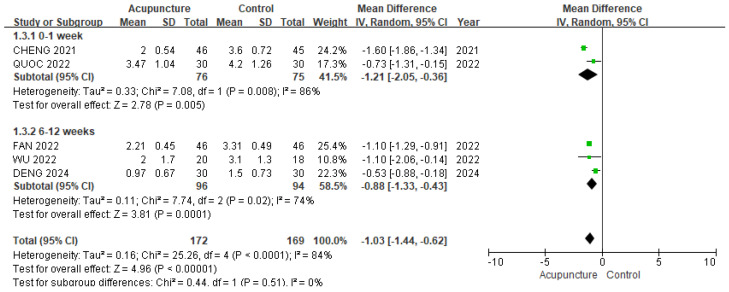
The forest plot of VAS (subgroup analyzed by duration of treatment prior to evaluation). The green squares represent the mean differences calculated for each study, and the horizontal lines indicate their 95% confidence intervals. The diamond represents the pooled mean difference with its 95% confidence interval [9,11,12,19,21].

**Figure 4 healthcare-13-02883-f004:**
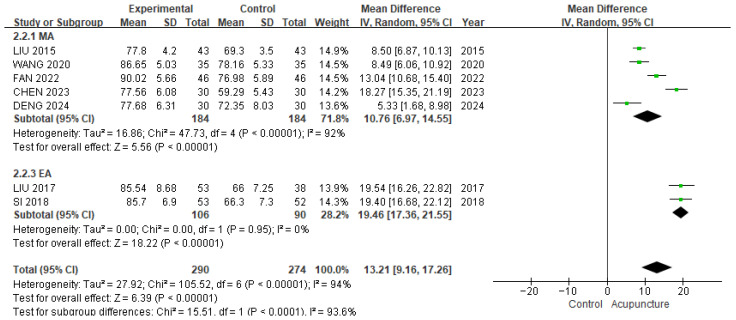
The forest plot of HSS score (subgroup analyzed by types of acupuncture). The green squares represent the mean differences calculated for each study, and the horizontal lines indicate their 95% confidence intervals. The diamond represents the pooled mean difference with its 95% confidence interval [19,24,25,30,33,35,36]. MA: manual acupuncture; EA: electroacupuncture.

**Figure 5 healthcare-13-02883-f005:**
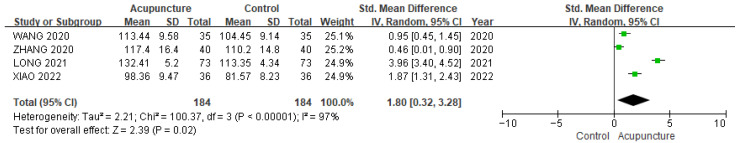
The forest plot of ROM of knee joint. The green squares represent the standardized mean differences calculated for each study, and the horizontal lines indicate their 95% confidence intervals. The diamond represents the pooled standardized mean difference with its 95% confidence interval [26,28,29,30].

**Figure 6 healthcare-13-02883-f006:**
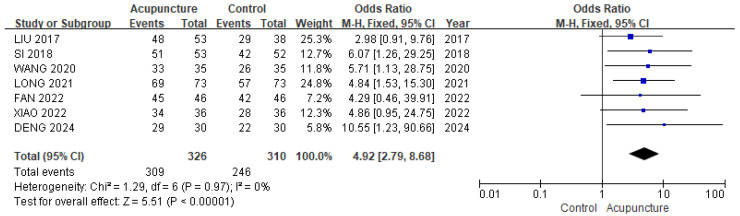
The forest plot of effective rate. The blue squares represent the odds ratios calculated for each study, and the horizontal lines indicate their 95% confidence intervals. The black diamond represents the pooled odds ratio with its 95% confidence interval [19,25,26,28,30,33,35].

**Figure 7 healthcare-13-02883-f007:**
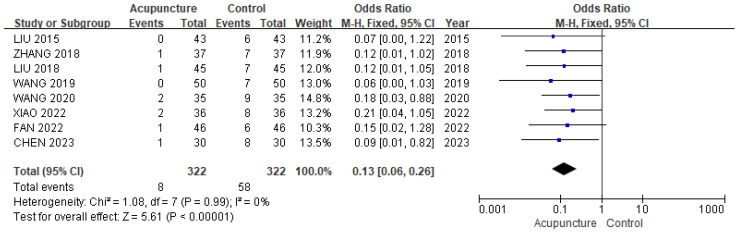
The forest plot of incidence of complications after surgery. The blue squares represent the odds ratios calculated for each study, and the horizontal lines indicate their 95% confidence intervals. The black diamond represents the pooled odds ratio with its 95% confidence interval [24,25,26,30,31,32,34,36].

**Table 1 healthcare-13-02883-t001:** Frequently used acupoints.

Acupuncture Point	GB34	EX-LE5	SP9	ST34	ST37	BL40, EX-LE2	SP10
Number of studies used	13	12	11	10	9	8	6

**Table 2 healthcare-13-02883-t002:** Summary of randomized controlled trials of acupuncture for tibial fracture.

Author (Year)	Gender (M/F)	Age	Intervention Group	Control Group	**Main Outcomes**
Deng (2024) [19]	I: 12/18 C: 14/16	I: 48.47 ± 11.36 C: 45.80 ± 14.29	MA (PC6, BL40, EX-LE5, SP10 (Fracture side), GV20, BL17, LR3, Ashi points, 1 time a day within 2 weeks after surgery; both sides SP10, BL17, SP9, GB39, BL11, 3 times a week for 3–4 weeks after surgery; ST36, GB39, BL17, BL25, GB34, BL23, BL11, 2 times a week for 1–3 month after surgery; 30 min; twirling manipulation, reinforcing–reducing manipulation) + basic rehabilitation	Basic rehabilitation	(1) VAS (2) HSS score (3) Effective rate
Chen (2023) [24]	I: 18/12 C: 19/11	I: 38.77 ± 3.55 C: 37.52 ± 3.29	MA (ST37, ST39, SP9, GB34, EX-LE5, BL40, EX-LE2, ST34; 1 time a day for 1 month, 30 min) +basic rehabilitation	Basic rehabilitation	(1) HSS score (2) Incidence of complications after surgery
Wu (2022) [18]	I: 13/7 C: 9/9	I: 43.1 ± 9.6 C: 42.5 ± 8.2	TEAS (SP10, GB33, SP9, ST35; 5 times a week for 6 weeks, 30 min, 2 Hz) + basic rehabilitation	Basic rehabilitation	(1) VAS
Quoc (2022) [9]	I: 19/11 C: 18/12	I: 37.0 ± 22.1 C: 35.6 ± 18.8	EA (ST36, SP8, SP9, GB34, SP6, SP10; 1 time a day for 7 days; needling duration for each treatment was not presented) +Medication (Paracetamol 1 g and Ketorolac 30 mg)	Basic rehabilitation + Medication (Paracetamol 1 g and Ketorolac 30 mg)	(1) VAS
Xiao (2022) [26]	I: 25/11 C: 23/13	I: 44.25 ± 7.59 C: 45.39 ± 7.21	MA (EX-LE5, BL40, ST34, ST37, EX-LE2, GB34, SP9; from 3 weeks after surgery, 1 time a day for 8 weeks, 30min; technique for receiving de qi) + basic rehabilitation		(1) ROM of knee joint (2) Effective rate (3) Incidence of complications after surgery
Fan (2022) [25]	I: 29/17 C: 28/18	I: 37.71 ± 8.89 C: 36.98 ± 8.89	MA (GB34, EX-LE2, SP9, EX-LE5, ST37, ST34, Ashi points; 1 time a week for within 8 weeks after surgery, 3 times a week for 9th–16th week after surgery, 1 time a week for 17th–24th week after surgery, 30min; technique for receiving de qi) + Basic rehabilitation + TCM fumigation and washing (1 time a day for 24 weeks, 30min)	Basic rehabilitation + TCM fumigation and washing (1 time a day for 24 weeks, 30 min)	(1) VAS (2) HSS score (3) Effective rate (4) Incidence of complications after surgery
Long (2021) [28]	I: not presented C: not presented	I: 44 ± 5 C: 42 ± 5	MA (GB39, BL11, BL17, non-fractured side Zhoufeng point; 1 time a day for 6–8 weeks, 30 min, acupuncture with joint mobilization) + basic rehabilitation	Basic rehabilitation	(1) ROM of knee joint (2) Effective rate
Cheng (2021) [27]	I: 26/20 C: 23/22	I: 41.83 ± 7.24 C: 41.91 ± 7.32	MA (GV20, fractured side LR3, SP10, both sides PC6; treatment frequency was not presented within 1 week after surgery; 30 min; twirling manipulation, reinforcing–reducing manipulation) + basic rehabilitation	Basic rehabilitation	(1) VAS
Zhang (2020) [29]	I: 25/15 C: 21/19	I: 42.8 ± 11.2 C: 40.5 ± 13.0	MA (ST34, EX-LE5, EX-LE2, SP9, GB34, BL40, ST37; 1 time a day for 30 days to 6 months after surgery, 30 min; technique for receiving de qi, reinforcing–reducing manipulation) + basic rehabilitation	Basic rehabilitation	(1) ROM of knee joint
Wang (2020) [30]	I: 20/15 C: 20/15	I: 52.91 ± 11.51 C: 52.87 ± 11.53	MA (EX-LE5, ST34, BL40, GB34, SP9, ST37, EX-LE2; 1 time a day for 3 months, 30 min; reinforcing–reducing manipulation) + basic rehabilitation	Basic rehabilitation	(1) HSS score (2) ROM of knee joint (3) Effective rate (4) Incidence of complications after surgery
Wang (2019) [31]	I: 21/29 C: 24/26	I: 42.5 ± 12.6 C: 42.4 ± 12.5	MA (ST34, EX-LE5, ST37, GB34; treatment frequency was not presented, 28 days to 3 months after surgery; 30 min) + basic rehabilitation	Basic rehabilitation	(1) Incidence of complications after surgery
Zhang (2018) [32]	I: 20/17 C: 19/18	I: 49. 84 ± 2. 94 C: 49. 73 ± 2. 57	MA (ST34, ST37, EX-LE5, BL40, SP9, GB34, EX-LE2; 1 time a day; the total period of treatment was not presented, 20–30 min; technique for receiving de qi, reinforcing–reducing manipulation) + basic rehabilitation	Basic rehabilitation	(1) Incidence of complications after surgery
Liu (2018) [34]	I: 30/15 C: 29/16	I: 47.9 ± 1.7 C: 46.3 ± 1.5	MA (EX-LE5, ST34, BL40, ST37, EX-LE2, SP9, GB34, treatment frequency was not presented, the total period of treatment was not presented; 30 min) + basic rehabilitation	Basic rehabilitation	(1) Incidence of complications after surgery
Si (2018) [33]	I: 38/15 C: 35/17	I: 37.2 ± 5.9 C: 36.8 ± 5.4	EA (SP10, GB34, LI4, LI11, SP6, Ashi points, within 7 days after surgery; ST40, GB34, BL17, 8 to 28 days after surgery; EX-LE5, both sides BL11, BL18, BL20, BL23, ST36, 29 to 56 days after surgery; treatment frequency was not presented; 30 min; twirling manipulation, reinforcing–reducing manipulation) + basic rehabilitation	Basic rehabilitation	(1) HSS score (2) Effective rate
Liu (2017) [35]	I: 32/21 C: 21/17	I: 38.11 ± 8.2 C: 36.36 ± 7.14	EA (Fracture side SP10, GB34, LI4, LI11, SP36, Ashi points, within 7 days after surgery; fractured side GB34, ST40, BL17, 8–28 days after surgery; both sides BL11, BL18, BL23, BL20, ST36, EX-LE5, ST34, 29–56 days after surgery; treatment frequency was not presented; 30 min; technique for receiving de qi, twirling manipulation, reinforcing–reducing manipulation) + basic rehabilitation	Basic rehabilitation	(1) HSS score (2) Effective rate
Liu (2015) [36]	I: 22/21 C: 23/20	I: 42.3 ± 12.4 C: 41.4 ± 11.8	MA (ST34, EX-LE5, ST37, BL40, EX-LE2, GB34, SP9; treatment frequency was not presented, the total period of treatment was not presented, from 20 days after surgery; 30 min) + basic rehabilitation	Basic rehabilitation	(1) HSS score (2) Incidence of complications after surgery

M: male; F: female; I: intervention group; C: control group; MA: manual acupuncture; EA: electroacupuncture; TEAS: transcutaneous electrical acupoint stimulation; TCM: traditional Chinese medicine; VAS: visual analogue scale; HSS: Hospital for Special Surgery; ROM: range of motion.

**Table 3 healthcare-13-02883-t003:** Summary of findings for GRADE ratings.

	Anticipated Absolute Effects (95% CI)	Relative Effect (95% CI)	Number of Participants (Studies)	Certainty of the Evidence (GRADE)	Rated Down Reasons
VAS	MD 1.03 lower (1.44 lower to 0.62 lower)	-	341 (5 RCTs)	⨁⨁◯◯ Low	Risk of bias Inconsistency
HSS score	MD 13.21 higher (9.16 higher to 17.26 higher)	-	564 (7 RCTs)	⨁⨁◯◯ Low	Risk of bias Inconsistency
ROM of knee joint	SMD 1.8 SD higher (0.32 higher to 3.28 higher)	-	368 (4 RCTs)	⨁◯◯◯ Very low	Risk of bias Inconsistency
Effective rate	156 more per 1000 (from 121 more to 177 more)	OR 4.92 (2.79 to 8.68)	636 (7 RCTs)	⨁⨁⨁◯ Moderate	Risk of bias
Incidence of complications after surgery	152 fewer per 1000 (from 167 fewer to 126 fewer)	OR 0.13 (0.06 to 0.26)	644 (8 RCTs)	⨁⨁⨁◯ Moderate	Risk of bias

⨁⨁⨁◯: Moderate certainty; ⨁⨁◯◯: Low certainty; ⨁◯◯◯: Very low certainty. CI: confidence interval; MD: mean difference; OR: odds ratio; SMD: standardized mean difference; VAS: visual analogue scale; HSS: Hospital for Special Surgery; ROM: range of motion.

## Data Availability

No new data were created or analyzed in this study. Data sharing is not applicable to this article.

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
