# Peer review of "Acupuncture for Post-Operative Pain Relief and Functional Improvement in Tibial Fracture: A Systematic Review and Meta-Analysis"

_healthcare, 2025, doi:10.3390/healthcare13222883_

Round 1
Reviewer 1 Report
Comments and Suggestions for Authors
According to careful consideration of the manuscript by Coo et al., it is evident that this work represents a systematic review and meta-analysis that is both valuable and of interest to readers. The authors have demonstrated commendable writing quality and have conducted a well-designed study. Therefore, I believe that this manuscript has the potential to be accepted for publication. However, it is recommended that the authors first provide responses to my suggestions and address the points raised.
1. In the Introduction section, additional details on acupuncture should be included about 1 paragraph, such as its characteristics, mechanisms of action, and relevant previous research studies.
2. In the manuscript, both the terms tribia fracture and tribial fracture are used. Consistency should be ensured throughout the manuscript.
3. In the Materials and Methods section, subsection 2.2 Search Strategy, the time frame or publication years included in the study should be clearly specified (i.e., from which year to which year).
4. In the Discussion section, the writing lacks fluency and is divided into an excessive number of paragraphs. The authors are encouraged to revise and improve the coherence and overall flow of the discussion.
Reviewer 2 Report
Comments and Suggestions for Authors
Thank you for the opportunity to review the manuscript entitled “Acupuncture for Post-operative Pain Relief and Functional Improvement in Tibial Fracture: A Systematic Review and Meta-analysis.”
This study is the first systematic review and meta-analysis focusing on acupuncture after tibial fracture surgery. It is methodologically rigorous, following PROSPERO registration and PRISMA guidelines, and includes a comprehensive search across multiple databases without language restrictions. The analysis covers important clinical outcomes like pain, functional recovery, range of motion, effectiveness, and complications, based on 16 randomized controlled trials involving 1,315 patients. The study also provides a balanced assessment of safety by reporting mild adverse events.
The included trials, mostly from China, have limited generalizability and often suffer from poor reporting of randomization, allocation concealment, and blinding, leading to a high risk of bias, especially since blinding is difficult in acupuncture studies. Many studies lacked key treatment and outcome details, reducing reliability. There was substantial heterogeneity in key outcomes like pain, function, and range of motion, with subgroup analyses only partially explaining it, which weakens confidence in combined results. According to GRADE, evidence quality was low or very low for most outcomes except for moderate certainty in complications and effective rate. The use of "effective rate" as an outcome is subjective and not widely accepted internationally, and important outcomes such as bone healing time or long-term function were not assessed. The study’s conclusion that acupuncture is relatively safe and effective overstates the evidence, and clinical recommendations should be more cautious. Additionally, reporting was sometimes unclear or inconsistent, with dense tables and insufficient explanation of heterogeneity figures.
Comments on the Quality of English LanguageThe English could be improved to more clearly express the research. The grammar and vocabulary are generally correct, but the writing style is sometimes awkward, with long and complex sentences that make the text harder to follow.
Reviewer 3 Report
Comments and Suggestions for Authors
- Line 15: I recommend revising into an impersonal and scientific style. This will improve the overall academic tone of the manuscript. The same suggestion applies to the rest of the text.
- Line 33: The introduction currently provides epidemiological data and patient age distributions. While useful, this may shift the reader’s attention away from the main focus, which is acupuncture as an intervention for pain relief. It might strengthen the introduction to instead expand on the background of acupuncture or to describe more clearly the nature of post-operative pain after tibial fracture.
- Line 53: Please revise into impersonal style. Additionally, I encourage the authors to explicitly state the importance of conducting this meta-analysis and to clarify what specific contributions it brings to the research field, compared to previous reviews.
- Line 60: Again, an impersonal writing style would be more appropriate for a scientific article. I will not repeat this comment, but I encourage the authors to revise the rest of the manuscript accordingly.
- Line 66: I kindly suggest uploading a completed PRISMA 2020 checklist as supplementary material. It would be helpful if the authors indicate in the checklist where each item is addressed in the manuscript.
- Line 104: Could the authors clarify whether any software tools were used for literature search management, study screening, or data extraction? If yes, please specify and provide references.
- Line 185: It would strengthen the Methods section to include a description of the acupuncture points most frequently used in the included studies, with appropriate references to support this information.
- Line 318: Please avoid mixing incidence and odds, as these are not equivalent statistical concepts. A careful review of the terminology used here would improve clarity.
- Line 345: At present, the manuscript does not include a sensitivity analysis. Such an analysis would be very valuable to assess how robust the findings are. For example:
- Are the results mainly driven by the largest studies?
- How does the high heterogeneity affect the strength of the conclusions?
- Line 347: I suggest beginning the Discussion with the main findings of the meta-analysis. For instance, summarize the improvements observed with acupuncture in pain, function, and complications, and then reflect on possible explanations for these effects (e.g., mechanisms of action, clinical context).
- Throughout the Discussion: Instead of repeatedly using the phrase “one study”, it may improve readability and fluency to cite the authors directly (e.g., Author et al.).
- Discussion – Limitations: Please expand the Limitations section to discuss more explicitly the high heterogeneity observed in several outcomes, as well as the absence of sensitivity analyses. Both aspects are important for readers to interpret the strength and reliability of the findings.
The quality of the English language could be improved. I would recommend that the authors seek professional language editing support to ensure clarity, consistency, and readability throughout the text.
